# Investigation and Source Apportionment of Air Pollutants in a Large Oceangoing Ship during Voyage

**DOI:** 10.3390/ijerph16030389

**Published:** 2019-01-30

**Authors:** Qiang Wang, Daizhi An, Rubao Sun, Mingxing Su

**Affiliations:** Center of Disease Control and Prevention of Chinese People’s Liberation Army, Beijing 100071, China; andaizhi@163.com (D.A.); sunrubao@163.com (R.S.); sumingxing@163.com (M.S.)

**Keywords:** large oceangoing ship, air pollution, source apportionment

## Abstract

The aims of this study were to determine compartmental air pollution during navigation of a large oceangoing ship and to identify preliminarily the major pollution sources. During the voyage of a bulk carrier ship, air samples were collected at 18 selected sites using a stratified sampling method. The concentrations of 15 pollutants were determined using gas chromatography. Results showed the concentrations of these pollutants varied significantly among the sampling sites, indicating major pollution sources at or nearby those locations. Five common factors extracted using factor analysis explained 89.092% of the total variance. Multivariate linear regression analysis showed the contributions to air pollution of these five common factors, i.e., the volatilization of ship paint, volatilization of ship-based oil, cooking activities, high-temperature release of rubber components on the ship and daily use of chemical products, and the application of deodorant and insecticide, were 41.07%, 25.14%, 14.37%, 11.78%, and 7.63%, respectively. Three significant groups were determined using cluster analysis based on their similarity, i.e., high, medium, and low pollution of sampling sites. This study established that the air of the bulk carrier ship was heavily polluted, and that effective identification of pollution sources could provide a scientific basis for its control.

## 1. Introduction

Increasingly, modern oceangoing ships are constructed as closed vessels, which is a design that could result in poor natural ventilation of the indoor environment. Large oceangoing ships are generally equipped with air purification equipment; however, the confined compartments of activity of the crew, complicated pollution sources, and air recycling inevitably produce internal air pollution [1,2]. Previous studies have found that numerous organic pollutants, such as those released by diesel, gasoline, paint, and crew activities, can affect air quality during voyages of large oceangoing ships [3,4,5]. Additionally, the widespread use of nonmetallic materials in ships, e.g., plastics, fiberglass, adhesives, refrigerants, and fire extinguishers can also lead to the release of pollutants [6]. The emission of inorganic and volatile organic compounds from nonmetallic materials is one of the main sources of air pollution in ship compartments [7]. Because of extended voyage durations of large oceangoing ships, air pollutant concentrations within ship compartments are likely to increase [8]. More importantly, because of the long periods spent at sea working and living in confined compartments, the internal environment of a ship has a direct impact on the physical/mental health and work efficiency of its crew [9].

For effective pollution control and air quality management, it is necessary to identify the pollution sources and their quantitative contributions. The application of different multivariate statistical techniques can offer better understanding of indoor air quality. Principal component analysis/factor analysis can be used for qualitative determination of latent sources but not for direct source apportionment. Originally, dispersion models were used for source apportionment; however, because of their limitations, there has been increasing interest in the use of receptor models. Receptor models assess the contributions of various sources, i.e., the so-called “receptors,” based on observations at sampling sites [10]. Multiple linear regression on absolute principal component scores has the advantage of not requiring a priori knowledge of the number of sources and source characteristics, which makes it suitable for source apportionment studies [11]. This technique can determine quantitatively both the loading of each pollutant from each source and the contribution of that source to the total pollutant concentration. Thus, it constitutes a valuable tool for developing appropriate strategies for effective management of indoor air quality.

In this study, we collected air pollution data during the voyage of a bulk carrier ship. The main objectives were to classify indoor air pollution levels, identify the major factors affecting indoor air quality, explore the possible sources of pollutants, and assess the potential source contributions for air pollution variables and for the sampling sites. These objectives were realized using a combination of statistical techniques including cluster analysis, factor analysis, and multiple linear regression. The conclusions derived from this study could provide effective support for the control of indoor air pollution.

## 2. Materials and Methods

### 2.1. Sampling Site

Onboard air sampling was conducted at various locations during the voyage of a bulk carrier ship. The method of stratified sampling was used to select 18 sampling sites at random. These sampling sites, which covered a variety of ship functions, mainly included working and living areas (Table 1).

### 2.2. Sampling Method

The actual sampling point was located in the middle of each sampling site at breathing height (1.5 m). Sampling points close to air conditioning outlets were avoided. Each sampling site had one sampling point, and one duplicate sample was established for each. An activated Tenax-TA adsorption tube (Agilent Technologies, Inc., Bellevue, WA, USA) was connected to a sampling pump (QT-2A, Lianyixing, China). All instruments were calibrated before use. For sampling, the pump was operated with a flow rate of 0.5 L/min and each air sample was collected continuously over a 20-min period. Thus, the volume of each sample was 10 L. After sampling, both ends of the sampling tube were closed and the samples were stored at 4 °C. The temperature, humidity, and atmospheric pressure at each of the sampling sites were recorded, and the sampled volume was converted to the standard state according to the following equation:
(1)$$V_{0}=V_{i}\frac{T_{0}}{273t}\times \frac{P}{P_{0}}$$
where
*V*_0_—sample volume under the standard state, L;*V_i_*—sampling volume, i.e., the product of sampling flow and sampling time, L;*T*—sampling point temperature, °C;*T*_0_—absolute temperature at the standard state, 273 K;*P*—atmospheric pressure at the sampling point, kPa;*P*_0_—atmospheric pressure at the standard state, 101.3 kPa.

### 2.3. Detection Method

Using a gas chromatograph (GC7890A, Agilent Technologies, Inc., Bellevue, WA, USA), 15 chemical pollutants were detected: Benzene, toluene, chlorobenzene, ethylbenzene, p-xylene, o-xylene, octane, hexanal, tetrachlorobenzene, limonene, undecane, nonanal, dodecane, decanal, and tetradecane. The instrument precision was verified and found within the range of validity. The chromatographic configuration comprised a 50-m-long capillary column with a 0.32-mm inner diameter quartz column, 2 methyl polysiloxane coating, and 1.05-μm film thickness. The operating conditions were such that the programmed injector temperature was 250 °C and the detector temperature was 260 °C. The initial temperature of the column was 50 °C, which was maintained for 10 min and then increased to 250 °C at the rate of 5 °C/min. The split ratio was 1:40. A standard sample of volatile organic compounds was obtained from the Chinese National Standard Substance Research Center.

### 2.4. Statistical Analysis

#### 2.4.1. Factor Analysis

Factor analysis is a technique for reducing the dimension of a variable and it can be used for classifying a variable. Its purpose is to use a limited number of invisible hidden variables to explain the relationship between the original variables. It can extract less, non-correlated, abstract, and comprehensive indices from more than one measured original variable a called varifactor (VF). Each of the original variables can be represented by a linear combination of these extracted common factors. In addition, in accordance with the influence of each factor on the original variables, the original variables can be divided into a number of classes equal to the number of factors. The theoretical model for factor analysis is
(2)$$z_{ji}=a_{f1}f_{1i}+a_{f2}f_{2i}+a_{f3}f_{3i}+\ldots +a_{fm}f_{mi}+e_{f}{}_{i}$$
where
*z_ji_*—the standardized score of variable *j*;*a*—factor loading;*f*—factor score;*m*—number of common factors used for variables;*e*—remainder of the error;*i*—sample quantity;*j*—number of variables.

#### 2.4.2. Multivariate Linear Regression

Multivariate linear regression is a statistical analysis method used to determine the quantitative relationship between two or more variables using regression analysis in mathematical statistics. It can be predicted or estimated based on an optimal combination of multiple independent variables. The multivariate linear regression model is
(3)$$Y=\beta_{0}+\beta_{1}X_{1i}+\beta_{2}X_{2i}+\ldots +\beta_{k}X_{ki}+\mu_{i}$$
where
*k*—number of explanatory variables;*β_K_*—partial regression coefficient;*β*_0_—constant;*X*—independent variable;*µ*—random error.

Although factor analysis can reduce the number of variables and simplify the problem, it cannot determine the absolute contributions of various emission sources. The independent factors extracted by factor analysis represent different sources, with the factor score as the variable of the multiple linear regression. Nevertheless, the contribution of the emission source to the receptor can be calculated [12].

#### 2.4.3. Cluster Analysis

Cluster analysis is a method for grouping objects into classes based on similarity within and between classes. The results of cluster analysis can help both in the interpretation of the data and in indicating patterns [13]. In hierarchical clustering, clusters are formed sequentially by starting with the most similar pair of objects and then forming subsequent higher clusters in a step-by-step process. In this study, hierarchical agglomerative cluster analysis was performed on the normalized dataset following Ward’s method using squared Euclidean distances as the measure of similarity [14].

## 3. Results and Discussion

### 3.1. Level of Pollutants at Ship Sampling Sites

Figure 1 shows the 15 chemical pollutants that were detected at the 18 sampling sites onboard the bulk carrier ship. According to Material Safety Data Sheets [15], some of the chemical pollutants are moderately and weakly toxic substances, and although they are not included in the Chinese National Standards, their potential impact on human health should be taken seriously [16,17,18]. Benzene, toluene, ethylbenzene, p-xylene, and o-xylene were detected at each sampling site. The xylene concentration at sampling site R was more than the maximum limit (30 mg/m^3^; GJB 11.2-1991). The concentrations of benzene and xylene at other sampling sites did not exceed the maximum limit, although they were found with high concentrations near the comfort concentration limit (0.8 and 4.5 mg/m^3^, respectively; GJB 11.2-1991), suggesting that long-term release of these substances could potentially be harmful to crew health.

### 3.2. Pollution Source Apportionment at the Sampling Sites

In accordance with the distribution of organic components at the different sampling sites, an initial analysis of air pollution sources at the various locations was undertaken using a receptor model based on the multivariate statistical method. First, all data were standardized with a mean of 0 and variance of 1. Before undertaking the factor analysis, a Kaiser-Meyer-Olkin (KMO) test and a Bartlett spherical test were performed. The KMO test, which is used to check the degree of correlation and partial correlation between variables, has values between 0 and 1. The closer the KMO statistic is to 1, the stronger the correlation between variables, the weaker the partial correlation, and the more effective the factor analysis. If the Bartlett spherical test determines that the correlation array is a unit array, then the independent factor analysis method for each variable is invalid. As shown in Table 2, the value of KMO was 0.307 and the Bartlett test showed significance (*p* < 0.05), indicating that the parameters of these samples were suitable for factor analysis.

The KMO criterion was adopted to decide the appropriate number of factors to be retained (i.e., only those factors with eigenvalues >1) [19]. An eigenvalue of 1 would mean that the component could explain the variability of one variable. Therefore, according to this criterion, a component should be capable of explaining the variability of at least one variable. In a very large dimensional dataset, including components capable of explaining the variability of only one variable might not be very useful. As shown in Table 3, five common factors were extracted according to the criterion of having a feature value >1. The five common factors listed in Table 3 explained 89.092% of the total variance (cumulative contribution rate reached 90%), indicating that the characteristics of the original air pollution data could be well represented. The results suggest five major sources of air pollution at the 18 sampling sites.

The rotating composition matrix was obtained though orthogonal rotation (Table 4). Factor loadings >0.75, between 0.5 and 0.75, and between 0.3 and 0.5 are considered strong, moderate, and weak, respectively [20]. Our results showed the first common factor had strong positive loadings on benzene, toluene, octane, tetrachlorobenzene, undecane, and decanal, and moderate positive loadings on p-xylene, o-xylene, and dodecane. It has been reported that benzene, toluene, ethylbenzene, p-xylene, o-xylene, and other volatile benzene compounds are the principal constituents of paint and coatings [21,22]. Octane, undecane, dodecane, and decane are also common components of paint and coatings [23]. Therefore, the first common factor might be explained by the volatilization of paint and coatings on the ship. The second common factor had strong positive loading on ethylbenzene and moderate positive loadings on p-xylene, o-xylene, hexanal, limonene, and nonanal. Ethylbenzene is often found as a component of gasoline, and the mixture of p-xylene, m-xylene, and o-xylene can be used as a gasoline additive [24]. Previous studies have found that gasoline and diesel steam mainly include aromatic hydrocarbon, isomeric alkene, naphthene, n-alkane, and other volatile organic compounds [25,26,27]. Moreover, limonene can be used as a dispersant of oil. Therefore, it was deduced that the second common factor was explained by the volatilization of shipboard oil. The third common factor had moderate positive loadings on toluene and hexanal and weak positive loading on nonylaldehyde. Studies have confirmed the exposure of cooks to benzene and toluene from biomass fuel combustion [28]. For example, Klein et al. [29] found that emissions from shallow frying, deep frying, and charbroiling comprise predominantly aldehydes of differing relative composition depending on the type of oil used. Therefore, it was inferred that the third common factor was caused by human activities such as cooking. The fourth common factor had strong positive loading on tetradecane and moderate positive loading on dodecane. Tetradecane can be used in rubber and organic synthesis of resin, and rubber is used widely onboard ships in applications including sealing parts of windows and doors, pipeline shock absorbers, bearings, and other mechanical parts. Dodecane is often used as the main raw material oil of many chemical products intended for daily use [30]. Thus, the fourth common factor could be explained by high-temperature release from rubber and the use of daily chemical products. The fifth common factor had strong positive loading on only chlorobenzene. Chlorobenzene is used widely in insect repellents, mildew treatment, and deodorization. It is the main component of the insecticide and deodorant used in the ship’s dump/toilet. Therefore, it is believed that the fifth common factor could be explained by the application of deodorant and insecticide.

### 3.3. Distribution of Sampling Site Pollution

Using factor analysis, the factor score of the common factor at each sampling point could be calculated. The factor score of each VF multiplied by its variance contribution rate extracts a common factor, which can then be weighted to obtain composite scores for each sampling site. For any sampling site, the larger the factor score, the more serious the pollution at that point [31]. It is seen from the scores that the pollution levels at the sampling points varied (Appendix A). Our results show that the overall pollution levels at each sampling site were in the order J > P > R > K > O > N > H > F > L > G > D > C > E > M > B > Q > I > A. For the first common factor VF1, the sampling point at site J had the highest score, showing the effect of volatilization of paint and coatings to be strongest, followed by R, O, K, H, F, and D. For the second common factor VF2, volatilization of shipboard oil at sampling site R was highest, followed in descending order by sampling sites P, K, O, G, and L. The third factor VF3 was clearly caused by human activities such as cooking at sampling site P, followed by J, N, F, M, and C. For the fourth common factor VF4, the most obvious effect from high-temperature release of rubber components and the daily use of chemical products was found at sampling site H, followed in descending order by P, R, and K. The fifth common factor VF5 had the highest score at sampling site N, mainly from the application of deodorant and insecticide, followed by R and H.

### 3.4. Contribution Rates of Sampling Site Pollution Source

Multiple linear regression can be performed using the factor score as the independent variable and the standardized concentration at each sampling point as the dependent variable. From Table 5, we have the regression equation Z = 0.366F1 + 0.224F2 + 0.128F3 + 0.105F4 + 0.068F5 + 5.805 × 10^−7^. It was determined that air pollution onboard the studied bulk carrier ship was affected by VF1, VF2, VF3, VF4, and VF5 and that their contribution rates were 41.07%, 25.14%, 14.37%, 11.78%, and 7.63%, respectively.

### 3.5. Classification of Sampling Site Pollution Levels

To implement control measures for the management of actual air pollution in ship compartments, in addition to factor analysis of the sources of compartment chemical pollutants, it is also necessary to develop a standard for the classification of the chemical pollutants. In this study, we further analyzed the spatial variations of air pollution and the degree of similarity between sampling sites using cluster analysis. The dendrogram obtained following Ward’s method, groups all 18 sampling sites into three statistically significant clusters (Figure 2). As in both Wang et al. [32] and Li et al. [33], the clustering procedure highlighted groups for which the sites had similar characteristics and natural source types. In our study, the hierarchical clustering analysis grouped the sampling sites into three clusters with similar air pollution characteristics. Cluster I (sampling sites A–I, K–O, and Q) corresponds to sampling sites with relatively weak pollution. Cluster II (sampling sites P and R) corresponds to sampling sites with moderate pollution Cluster III (sampling site J) corresponds to strong pollution and it represents a sampling site influenced primarily by the volatilization of paint and coatings and by human activities such as cooking. The extracted group information could be used to reduce the number of sampling sites in further study without significant loss of information.

## 4. Conclusions

The present study investigated the compartmental air pollution that occurred during the voyage of the large oceangoing ship. Factor analysis helped identify the factors/sources responsible for the variation of air pollution in different compartments. The VFs obtained from factor analysis indicated there were five common factors responsible for the variation of air pollution. Receptor modeling using multivariate linear regression analysis showed the contributions to air pollution of these five common factors, i.e., the volatilization of ship paint, volatilization of ship-based oil, cooking activities, high-temperature release of rubber components on the ship and daily use of chemical products, and the application of deodorant and insecticide, were 41.07%, 25.14%, 14.37%, 11.78%, and 7.63%, respectively. Furthermore, hierarchical clustering analysis grouped the sampling sites into three clusters with similar air pollution characteristics. The extracted group information could be used to reduce the number of sampling sites in further studies without significant loss of information. The findings of this study highlight the usefulness of multivariate statistical techniques in the identification and apportionment of pollution sources/factors, which can reveal crucial information regarding air pollution during the voyage of a large oceangoing ship.

### Limitations and Future Research

This study explored pollution sources only from the perspective of multivariate statistics. The methods adopted in this research should be verified further with other case studies. Some air pollution variables are known to receive contributions from unidentified sources; thus, further work will be necessary in order for them to be identified.

## Figures and Tables

**Figure 1 ijerph-16-00389-f001:**
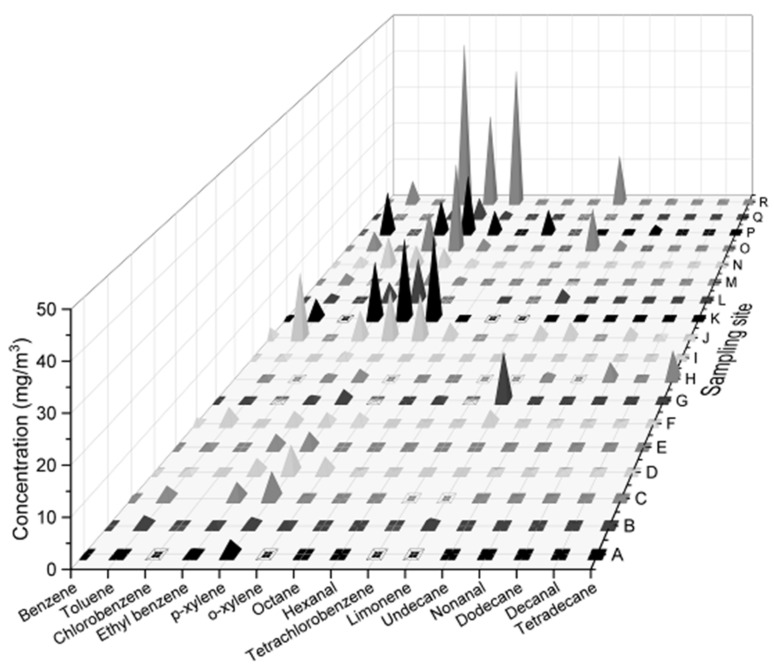
Average concentrations of 15 air pollutants at different sampling sites.

**Figure 2 ijerph-16-00389-f002:**
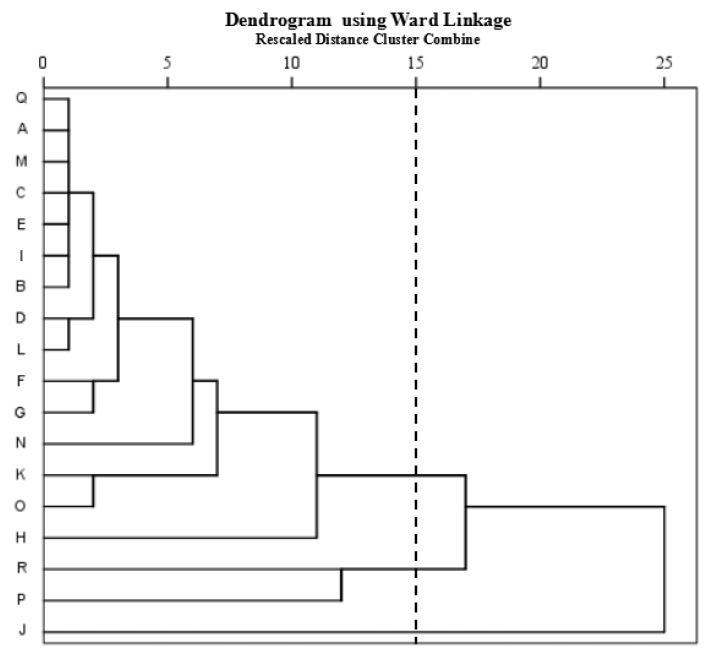
Dendrogram obtained using cluster analysis.

**Table 1 ijerph-16-00389-t001:** Identification of the 18 sampling sites onboard the bulk carrier ship.

Sampling Site	Location	Sampling Site	Location	Sampling Site	Location
A	Wheel house	G	Pastry room	M	Engine control room
B	Compartment 1	H	Incineration room	N	Shower room
C	Dining room	I	Switching room	O	Compartment 2
D	Alley way 1	J	Lavatory	P	Galley
E	Engine room	K	Store room	Q	Pump room
F	Cabin	L	Alley way 2	R	Fuel oil tank

**Table 2 ijerph-16-00389-t002:** Results of Kaiser–Meyer–Olkin (KMO) and Bartlett tests.

Kaiser–Meyer–Olkin measure of sampling adequacy		0.307
Bartlett’s test of sphericity	Approx. Chi-square	342.816
	df	105
	Sig.	0.000

**Table 3 ijerph-16-00389-t003:** Total variance explained.

Component	Initial Eigenvalues			Extraction Sums of Squared Loadings		
	Total	% Of Variance	Cumulative %	Total	% of Variance	Cumulative %
1	5.497	36.649	36.649	5.497	36.649	36.649
2	3.355	22.367	59.015	3.355	22.367	59.015
3	1.915	12.763	71.779	1.915	12.763	71.779
4	1.576	10.508	82.287	1.576	10.508	82.287
5	1.021	6.805	89.092	1.021	6.805	89.092
6	0.659	4.395	93.487			
7	0.553	3.686	97.173			
8	0.236	1.572	98.745			
9	0.091	0.605	99.350			
10	0.042	0.283	99.633			
11	0.024	0.162	99.795			
12	0.018	0.118	99.913			
13	0.011	0.071	99.985			
14	0.002	0.014	99.998			
15	0.000	0.002	100.000			

**Table 4 ijerph-16-00389-t004:** Component matrix.

Variables	Component				
	1	2	3	4	5
Benzene	0.804	−0.417	0.219	−0.237	0.023
Toluene	0.804	0.223	0.527	0.003	−0.010
Chlorobenzene	−0.125	−0.025	0.093	−0.141	0.948
Ethylbenzene	0.457	0.752	−0.412	0.035	0.133
p-xylene	0.515	0.698	−0.109	0.080	−0.076
o-xylene	0.561	0.647	−0.360	−0.001	0.102
Octane	0.862	−0.307	0.266	−0.205	−0.003
Hexanal	−0.085	0.562	0.709	0.390	−0.030
Tetrachlorobenzene	0.870	−0.133	0.164	−0.282	−0.003
Limonene	0.368	0.518	−0.377	−0.078	−0.174
Undecane	0.845	−0.217	0.228	0.165	−0.080
Nonanal	−0.131	0.748	0.496	0.227	0.122
Dodecane	0.613	−0.475	−0.155	0.601	0.087
Decanal	0.791	−0.059	−0.399	−0.024	0.144
Tetradecane	0.159	−0.356	−0.211	0.879	0.084

**Table 5 ijerph-16-00389-t005:** Regression coefficients of multiple regression equation.

Model	Non-Standardized Coefficient		Standardized Coefficient
	*B*	Standard Error	Beta
Constant	5.805 × 10^−7^	0.000	
F1	0.366	0.000	0.788
F2	0.224	0.000	0.481
F3	0.128	0.000	0.274
F4	0.105	0.000	0.226
F5	0.068	0.000	0.146

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
