# Peer review of "Investigation and Source Apportionment of Air Pollutants in a Large Oceangoing Ship during Voyage"

_ijerph, 2019, doi:10.3390/ijerph16030389_

Reviewer 1 Report

Comments:

Title: should be removed source apportionment, or please apply the most recent techniques such as PMF, and CMB for the dataset

Abstract: it’s not clear. It need to be revised and re-writed

                   1.       Introduction: it is very basic information. It should be expanded and supported by some new study. This introduction must be changed and  written in good way. The current format isn’t acceptable.

Page2, Line1-4: The aim of the study is not cleared showed.

                    2.       Methodology:

                It is written briefly, it should be written in scientific way and change sub sections numbers, for example put sampling site (2.3) instead section 2.1

3.       Results and discussion:

Table 1: add unit of concentration of species.

Page 8, Figure: add title of the figure

 4.       Conclusions:

 Please re-write the conclusion and removing all number points. Write clear conclusion, what have you found in this study? What is still a gap for future research?

Author Response

Response to Reviewer 1 Comments

Point 1: Title: should be removed source apportionment, or please apply the most recent techniques such as PMF, and CMB for the dataset. 

Response 1: We would like to thank the reviewer for these helpful suggestions. According to reviewers comment, source apportionment have been removed from the title.

Point 2: Abstract: it’s not clear. It need to be revised and re-writed.

Response 2: We have rewritten the abstract according to reviewers comment.

Point 3:  Introduction: it is very basic information. It should be expanded and supported by some new study. This introduction must be changed and written in good way. The current format isn’t acceptable.

Response 3: We would like to thank the reviewer for helpful suggestions. According to reviewers comment, we have rewritten the introduction. New references have been added in the revised manuscript.

Point 4: Page2, Line1-4: The aim of the study is not cleared showed.

Response 4: According to reviewers comment, we have deleted these sentences and the new content is as follows.

In this study, we collected air pollution data during the voyage of a bulk carrier ship. The main objectives were to classify indoor air pollution levels, identify the major factors affecting indoor air quality, explore the possible sources of pollutants, and assess the potential source contributions for air pollution variables and for the sampling sites. These objectives were realized using a combination of statistical techniques including cluster analysis, factor analysis, and multiple linear regression. The conclusions derived from this study could provide effective support for the control of indoor air pollution.

Point 5: Methodology: It is written briefly, it should be written in scientific way and change sub sections numbers, for example put sampling site (2.3) instead section 2.1

Response 5: Modification has been made according to reviewers comment. Some content of the methodology has been deleted, and sub section numbers have been changed accordingly.

Point 6: Results and discussion: Table 1: add unit of concentration of species.

Response 6: According to reviewers comment, we have added concentration unit on the axis of the Figure 1.

Point 7: Page 8, Figure: add title of the figure.

Response 7: We have added a title to each figure in the new manuscript.

Point 8: Conclusions: Please re-write the conclusion and removing all number points. Write clear conclusion, what have you found in this study? What is still a gap for future research?

Response 8: We have rewritten the conclusion according to reviewers comment. All number points have been removed. Limitations and future research has been added in the new manuscript. The modified sentences are as follows:

4. Conclusions

The present study investigated the compartmental air pollution that occurred during the voyage of the large oceangoing ship. Factor analysis helped identify the factors/sources responsible for the variation of air pollution in different compartments. The VFs obtained from factor analysis indicated there were five common factors responsible for the variation of air pollution. Receptor modeling using multivariate linear regression analysis showed the contributions to air pollution of these five common factors, i.e., the volatilization of ship paint, volatilization of ship-based oil, cooking activities, high-temperature release of rubber components on the ship and daily use of chemical products, and the application of deodorant and insecticide, were 41.07%, 25.14%, 14.37%, 11.78%, and 7.63%, respectively. Furthermore, hierarchical clustering analysis grouped the sampling sites into three clusters with similar air pollution characteristics. The extracted group information could be used to reduce the number of sampling sites in further study without significant loss of information. The findings of this study highlight the usefulness of multivariate statistical techniques in the identification and apportionment of pollution sources/factors, which can reveal crucial information regarding air pollution during the voyage of a large oceangoing ship.

5. Limitations and Future Research

This study explored pollution sources only from the perspective of multivariate statistics. The methods adopted in this research should be verified further with other case studies. Some air pollution variables are known to receive contributions from unidentified sources; thus, further work will be necessary for their identification. 

Reviewer 2 Report

Dear Editor and Authors, 

the manuscript presents the results of a source apportionment study in selected sites of a ship. The interesting part of the paper is the type of the indoor microenvironments studied. However, some important information from the methodology and results presentation is still missing. I would recommend a major revision of it before reconsidering of its publishing. Particularly:

the title of the manuscript is misleading: The source apportionment term refers to air pollution not air quality. You could refer specifically to the substances you measured e.g. VOCs or organic compounds or pollutants or ...but not air quality.

 do not use the word 'space' , you can use 'sampling site'

At the end of the Introduction, the scope of this manuscript should be clearly stated. Which is the innovative aspect of the present work? What is the added value to the literature? Why was performed?

At the beginning of the Methodology, the measures/detected substances should be presented. You describe the analysis without mentioning which pollutants you have analysed.

A detailed presentation of each site and its characteristics is missing (maybe in a Table). The reader does not know which are the differences between the sites. This will help also to the explanation of the model results. 

page 2, line 20: breathing zone, not belt

page 4, lines 1-4: support this information with literature references

Table 1 is problematic in reading. Format needs to be changed. Also, sampling sites coding is not adequate. Detailed description should have been given previously (see comment 5). Finally, check the significant digits for the concentration values.

please explain in more details why the 5-factor solution was selected. How many samples were inserted in the model? Was any quality check performed?

results: literature references are needed for supporting your scientific findings (e.g. corresponding of factors to sources)

limitations of the study should be briefly mentioned 

Author Response

Response to Reviewer 2 Comments

Point 1: The manuscript presents the results of a source apportionment study in selected sites of a ship. The interesting part of the paper is the type of the indoor microenvironments studied. However, some important information from the methodology and results presentation is still missing. I would recommend a major revision of it before reconsidering of its publishing. Particularly:

the title of the manuscript is misleading: The source apportionment term refers to air pollution not air quality. You could refer specifically to the substances you measured e.g. VOCs or organic compounds or pollutants or ...but not air quality. 

Response 1: We appreciate reviewers helpful suggestion. According to reviewers comment, air quality has been changed to air pollution.

Point 2: do not use the word 'space' , you can use 'sampling site'

Response 2: Modification has been made according to reviewers comment. The word space has been changed to sampling site in new manuscript.

Point 3: At the end of the Introduction, the scope of this manuscript should be clearly stated. Which is the innovative aspect of the present work? What is the added value to the literature? Why was performed?

Response 3: According to  reviewers comment, we  have rewritten the Introduction. The modified sentences  are as follows:

In this study, we collected air pollution data during the voyage of a bulk carrier ship. The main objectives were to classify indoor air pollution levels, identify the major factors affecting indoor air quality, explore the possible sources of pollutants, and assess the potential source contributions for air pollution variables and for the sampling sites. These objectives were realized using a combination of statistical techniques including cluster analysis, factor analysis, and multiple linear regression. The conclusions derived from this study could provide effective support for the control of indoor air pollution.

Point 4: At the beginning of the Methodology, the measures/detected substances should be presented. You describe the analysis without mentioning which pollutants you have analysed.

Response 4:  According to reviewers comment, the names of all the detected pollutants are listed in the method section 2.3.

Using a gas chromatograph (GC7890A, Agilent Technologies, Inc., Bellevue, WA, USA), 15 chemical pollutants were detected: benzene, toluene, chlorobenzene, ethylbenzene, p-xylene, o-xylene, octane, hexanal, tetrachlorobenzene, limonene, undecane, nonanal, dodecane, decanal, and tetradecane.  

Point 5: A detailed presentation of each site and its characteristics is missing (maybe in a Table). The reader does not know which are the differences between the sites. This will help also to the explanation of the model results.

Response 5: According to reviewers comment, information for each sample point has been added in Table 1.

Point 6: page 2, line 20: breathing zone, not belt.

Response 6: We have changed breathing belt to breathing zone.

Point 7: page 4, lines 1-4: support this information with literature references.

Response 7: According to reviewers comment, new references have been added in the revised manuscript. The additional references are as follows:

15. Material Safety Data Sheets. http://cheman.chemnet.com/en-msds.html

16. Gustafson, D.L.; Long, M.E.; Thomas, R.S.; Benjamin, S.A.; Yang, R.S. Comparative hepatocarcinogenicity of hexachlorobenzene, pentachlorobenzene, 1,2,4,5-tetrachlorobenzene, and 1,4-dichlorobenzene: application of a medium-term liver focus bioassay and molecular and cellular indices. Toxicol. Sci. 2000, 53(2), 245-252.

17. Vaghef, H.; Hellman, B. Demonstration of chlorobenzene-induced DNA damage in mouse lymphocytes using the single cell gel electrophoresis assay. Toxicology. 1995, 96(1), 19-28.

18. Saghir, S.A.; Zhang, F.; Rick, D.L.; Kan, L.;, Bus, J.S.; Bartels, M.J. In vitro metabolism and covalent binding of ethylbenzene to microsomal protein as a possible mechanism of ethylbenzene-induced mouse lung tumorigenesis. Regul. Toxicol. Pharmacol. 2010, 57(2-3), 129-135.

Point 8: Table 1 is problematic in reading. Format needs to be changed. Also, sampling sites coding is not adequate. Detailed description should have been given previously (see comment 5). Finally, check the significant digits for the concentration values.

Response 8: According to reviewers comment, we have converted the table into a graph (Figure 1). And information for each sample point has been added in Table 1.

Point 9: please explain in more details why the 5-factor solution was selected. How many samples were inserted in the model? Was any quality check performed?

Response 9: As we know, there are usually two ways to select components/factors. (1) The eigenvalue criterion; (2) The proportion of the variance explained criterion. In our study, we chose the method 1. An eigenvalue of one would mean that the component would explain about one variables worth of variability. So, according to this criterion, a component should at least explain one variables worth of variability. We can say that we will include only those eigenvalues whose value is greater than or equal to one. In a very large dimensional dataset including components capable of explaining only one variable may not be very useful. Moreover, as shown in Table 3, the five common factors explained 89.092% of the total variance. The cumulative contribution rate of the five common factors is almost 90%. So we chose the 5-factor solution. In our study, 18 samples were inserted in the model. Total number of samples tested was less than 100. Indeed, the reliability of factor analysis is closely related to the number of samples. However, scholars have no consistent conclusion on how many samples should be used for factor analysis to make the results most reliable. Most scholars agree that "factor analysis must have reliable results, and the number of samples tested is less than the number of items on the scale" (Wu MingLong. SPSS Statistical Application Practice: Questionnaire Analysis and Application Statistics. Science Press. 2003.).  For example, if there are 20 items in a component table, the number of samples must not be less than 20 for factor analysis. Factor analysis also needs to be tested for validity. In this study, we tested the KMO values to see if they were suitable for factor analysis.

Point 10: results: literature references are needed for supporting your scientific findings (e.g. corresponding of factors to sources).

Response 10: New references have been added in the revised manuscript. The references are as follows:

23. Schweitzer P.A. Paintings and Coatings: Applications and Corrosion Resistance. CRC. Press. 2006.

24. Heibati, B.; Godri, Pollitt, K.J.; Charati, J.Y.; Ducatman, A.; Shokrzadeh, M.; Karimi, A.; Mohammadyan, M. Biomonitoring-based exposure assessment of benzene, toluene, ethylbenzene and xylene among workers at petroleum distribution facilities. Ecotoxicol. Environ. Saf. 2018, 149, 19-25.

30. Förster, M.; Bolzinger, M.A.; Ach, D.; Montagnac, G.; Briançon, S. Ingredients tracking of cosmetic formulations in the skin: a confocal Raman microscopy investigation. Pharm. Res. 2011, 28(4), 858-872.

Point 11: limitations of the study should be briefly mentioned.

Response 11: The new manuscript have been added the section of limitations and future research.

5. Limitations and Future Research

This study explored pollution sources only from the perspective of multivariate statistics. The methods adopted in this research should be verified further with other case studies. Some air pollution variables are known to receive contributions from unidentified sources; thus, further work will be necessary for their identification.

Reviewer 3 Report

Can be found in the attached pdf-file.

Author Response

Response to Reviewer 3 Comments

Point 1: Major comments

This paper needs a comprehensive review of the written English language. The language quality in the current condition of the manuscript is too poor to allow publication. I will not comment the language in specifics here, but I highly recommend that a native English speaking person take a look at the grammar here.

Response 1: In light of this comment, the manuscript has been reviewed by an English language editing service (Edanz Group Ltd.). Sentences containing grammatical and/or spelling mistakes have been corrected. We believe that the revised manuscript clearly presents our findings to the reader.

Point 2: The introduction is of poor quality and more background is needed for the reader to get acquainted with the research field and previous scientific accomplishments.

Response 2: We would like to thank the reviewer for helpful suggestions. According to reviewers comment, we have rewritten the introduction.

Point 3: Several of the tables and figures are unnecessary to present and can be attached into an appendix instead. I will recommend a major revision before this manuscript can be considered to get published.

Response 3: According to reviewers comment, figures 1 and 2 have been removed and table 5 has been added to the appendix.

Point 4: Minor comments

P1.

L. 28-29. Please delete “development of science and technology…”.

Response 4: We have deleted “development of science and technology…” according to reviewers comment.

Point 5: L. 40. What is “the old problem”?

Response 5: We apologize for the confusion caused by these words. We have rewritten the introduction and deleted the old problem in the revised manuscript.

Point 6: L. 43. Please develop the sentence: “…widespread use of new materials…” etc. What do you mean?

Response 6: According to reviewers comment, we have rewritten and developed the sentence.

Additionally, the widespread use of nonmetallic materials in ships, e.g., plastics, fiberglass, adhesives, refrigerants, and fire extinguishers can also lead to the release of pollutants [6]. The emission of inorganic and volatile organic compounds from nonmetallic materials is one of the main sources of air pollution in ship compartments [7].

Point 7: P2.

L. 14-18. Please state the type of ship you studied. I imagine that all ships are different and hence have different prerequisites for indoor air pollution.

Response 7: The type of the ship we studied is the bulk carrier ship. According to reviewers comment, we have stated the type of the ship in the Materials and Methods.

2.1. Sampling Site

Onboard air sampling was conducted at various locations during the voyage of a bulk carrier ship.

Point 8: L. 15. Please explain what you mean with stratified sampling.

Response 8: First, we divided the entire compartments into living and working compartments. We then selected a random sample based on the ratio of the two compartments. In this way, in the same total sample size, we can produce a smaller error estimate than the random sampling method. This is the stratified sampling we use.

Point 9: L. 22. With parallel sampling you mean that you were taking duplicates?

Response 9: Yes, we mean duplicate samples. In order to avoid misunderstanding, we have changed parallel samples to duplicate samples” in the revised manuscript.

Point 10: L. 24. “That pump”. Which pump? Explain.

Response 10:  That pump means the sampling pump. We have changed That pump to the pump” in the revised manuscript.

Point 11: L. 26-27. “Light was avoided in the test”. What light and why? What did you test?

Response 11: We apologize for the confusion caused by the wrong sentence. We have changed light was avoided in the test to the samples were stored at 4 °C .

Point 12: L. 27. By “sample” you mean sampling site?

Response 12: Yes, we mean sampling site. We have changed sample to “sampling sites” in the revised manuscript.

Point 13: L. 39. “The instrument was in the verification period before use”. What does this mean?

Response 13: Sorry for the confusion caused by the wrong sentence. The sentence has been change to The instrument precision was verified and found within the range of validity. .

Point 14: P3.

L. 24. What is X and µ in this formula?

Response 14: We have added what X and µ represent in the formula.

Point 15: P4.

L. 1-10. You must state in some table where these spaces with associated letters are located. In the current state it is impossible to follow what you mean.

Response 15: According to reviewers comment, we have added Figure 1, which provides information about the sampling sites.

Point 16: L. 1-10. State the limits you are referring to.

Response 16: We have added the standard limits into new manuscript.

Point 17: L. 8-9. This last sentence doesn’t say anything. Remove.

Response 17: We have deleted the sentence according to reviewers comment.

Point 18: P5.

Table 1. It would been a lot better if you took these numbers and put them into a figure.

Response 18: According to reviewers comment, we have converted the table into a figure.

Point 19: Table 1. Again, these letters don’t say anything. Improve.

Response 19: According to reviewers comment, we have converted the table into a figure.

The figure does present the results more visually.

Point 20: P7.

L. 1-9. Explain the Kaiser-Meyer-Olkin and Bartlett spherical tests thoroughly here. Current explanation is way to poor.

Response 20: We have explained the application of these two methods in the section of the Results and Discussion.

Before undertaking the factor analysis, a Kaiser–Meyer–Olkin (KMO) test and a Bartlett spherical test were performed. The KMO test, which is used to check the degree of correlation and partial correlation between variables, has values between 0 and 1. The closer the KMO statistic is to 1, the stronger the correlation between variables, the weaker the partial correlation, and the more effective the factor analysis. If the Bartlett spherical test determines that the correlation array is a unit array, then the independent factor analysis method for each variable is invalid.

Point 21: L. 11. Figure 1t?

Response 21: We apologize for the mistake in our handwriting. And we have deleted the figure in new manuscript.

Point 22: Table 3. Explain why you choose five factors instead of 15 which would explain 100% of the variance?

Response 22: As we know, there are usually two ways to select components/factors. (1) The eigenvalue criterion; (2) The proportion of the variance explained criterion. In our study, we chose the method 1. An eigenvalue of one would mean that the component would explain about one variables worth of variability. So, according to this criterion, a component should at least explain one variables worth of variability. We can say that we will include only those Eigenvalues whose value is greater than or equal to one. In a very large dimensional dataset including components capable of explaining only one variable may not be very useful. Moreover, as shown in Table 3, the five common factors explained 89.092% of the total variance. The cumulative contribution rate of the five common factors is almost 90%. So we chose 5 factors instead of 15.

Point 23: P8.

Figure 1. This figure doesn’t say anything. Remove or explain it thoroughly.

Response 23: We have deleted the figure according to reviewers comment.

Point 24: L. 3. “According to the study”. What study? This is not the way to refer.

Response 24: We have deleted the According to the study in new manuscript.

Point 25: P10.

Figure 2. Remove. You are working and presenting data from a 5D-component matrix. Why are you then showing a 3D-component matrix plot? I don’t understand.

Response 25: We have deleted the figure according to reviewers comment.

Point 26: Table 5. These data could go to an appendix.

Response 26: We have transferred these data to the appendix.

Point 27: P. 12.

L. 18. You are using “space” throughout the manuscript. However, I think it would be better to use “indoor environments” or “compartments”.

Response 27: We have changed space to compartments according to reviewers comment.

Point 28: L. 20. “..relatively high in spaces.”. Relative to what?

Response 28: We mean relative to the standard limits.  In order to avoid confusion, we have deleted the sentence The concentrations of benzene, toluene, ethylbenzene, p-xylene and o-xylene were relatively high in spaces.

Round  2

Reviewer 2 Report

all of my comments have been adequately addressed, so I recommend the publication of the paper. Just a correction regarding the title: I did not recommend to delete the term 'source apportionment' from the title. Instead, I would recommend the title: 

Investigation and source apportionment of air pollution in a large oceangoing ship during voyage

Reviewer 3 Report

The paper has improved sufficiently and is ready for publication.